# Renal survival and treatment of adult patients with Primary Focal Segmental glomerulosclerosis: A historical cohort study of the National Greek Registry

Smaragdi Marinaki[1], Panagiotis Kompotiatis[1]*, Ioannis Michelakis[1]*, Maria Stangou[2], Aikaterini Papagianni[2], Maria Koukoulaki[3], Synodi Zerbala[3], Dimitrios Xydakis[4], Nikolaos Kaperonis[5], Evangelia Dounousi[6], Spyridon Golfinopoulos[7], Ioannis Stefanidis[7], Aggeliki Paikopoulou[8], George Moustakas[9], Kostas Stylianou[10], Ioannis Tzanakis[11], Marios Papasotiriou[12], Dimitrios Goumenos[12], Aimilios Andrikos[13], Pelagia Kriki[14], Stylianos Panagoutsos[14], Eva Kiousi[15], Eirini Grapsa[15], Georgios Koutroumpas[16], Panagiotis Pateinakis[17], Dorothea Papadopoulou[17], Vasilios Liakopoulos[18], Dimitra Bacharaki[19], Penelope Kouki[20], Dimitrios Petras[20], Gerasimos Bamichas[21], Ioannis Boletis[1]

1 Department of Nephrology, Laiko General Hospital, National and Kapodistrian University, Athens, Greece, 2 Department of Nephrology, Hippokration General Hospital, Aristotle University, Thessaloniki, Greece, 3 Department of Nephrology, General Hospital of Nikaia, Piraeus, Greece, 4 Department of Nephrology, Venizelio General Hospital of Heraklion, Heraklion Crete, Greece, 5 Department of Nephrology, Hellenic Red Cross Hospital Korgialeneio-Benakeio, Greece, 6 Department of Nephrology, University Hospital of Ioannina, Ioannina, Greece, 7 Department of Nephrology, University Hospital of Larissa, Larissa, Greece, 8 Department of Nephrology, Evangelismos General Hospital, Athens, Greece, 9 Department of Nephrology, Gennimatas General Hospital of Athens, Athens, Greece, 10 Department of Nephrology, University Hospital of Heraklion, Heraklion Crete, Greece, 11 Department of Nephrology, General Hospital of Chania, Chania Crete, Greece, 12 Department of Nephrology, University Hospital of Patras, Patras, Greece, 13 Department of Nephrology, Hatzikosta General Hospital of Ioannina, Ioannina, Greece, 14 Department of Nephrology, University Hospital of Alexandroupolis, Alexandroupoli, Greece, 15 Department of Nephrology, Aretaieio Hospital, National and Kapodistrian University of Athens, Athens, Greece, 16 Department of Nephrology, General Hospital of Volos, Volos, Greece, 17 Department of Nephrology, Papageorgiou General Hospital of Thessaloniki, Thessaloniki, Greece, 18 Section of Nephrology, 1st Department of Medicine, AHEPA University General Hospital, Thessaloniki, Greece, 19 Department of Nephrology, Attikon University Hospital, National and Kapodistrian University, Athens, Greece, 20 Department of Nephrology, Hippokration General Hospital, Athens, Greece, 21 Department of Nephrology, Papanikolaou General Hospital of Thessaloniki, Thessaloniki, Greece

* pkompotiatis@gmail.com (PK); michgiannis@gmail.com (IM)

**Data Availability Statement:** All data relevant to the study are included in the article.

## Abstract

### Background/Objective

Primary Focal and Segmental glomerulosclerosis (FSGS) is one of the most common causes of idiopathic nephrotic syndrome. Our aim was to describe a large cohort of patients with primary FSGS, identify risk factors associated with worse renal survival and assess the impact of different immunosuppressive regiments on renal survival.

**Funding:** The author(s) received no specific funding for this work.

**Competing interests:** The authors have declared that no competing interests exist.

## Methods

This was a historical cohort study of adults who were diagnosed with primary FSGS from March 26, 1982, to September 16, 2020. The primary outcome was progression to ESRD.

## Results

We included 579 patients. The mean age was 46 (±15) years of age, with 378 (65%) males and median 24-hour proteinuria was 3.8 (2–6) g. In multivariable analysis only eGFR (HR: 0.97 per ml/min increase, 95% CIs 0.95–0.98) and remission status (complete remission (HR: 0.03, 95% CIs 0.003–0.22) and partial remission (HR: 0.28, 95% CIs 0.13–0.61) compared to no remission) were associated with renal survival. Among patients who received immunosuppression compared to those that did not, there was a higher percentage of complete remission (121 (41%) vs. 40 (24%), p<0.001), and higher percentage of relapses (135 (64%) vs. 27 (33%), p<0.001). Immunosuppression and its type (glucocorticoids vs. cyclosporine ± glucocorticoids) were not associated with renal survival.

## Conclusion

In primary FSGS, complete and partial remission were associated with improved renal survival. Further randomized studies are needed to assess the efficacy of different therapeutic agents and guide treatment.

## Introduction

Focal Segmental Glomerulosclerosis (FSGS) is considered one of the most common causes of nephrotic syndrome [1]. The etiology of FSGS is diverse and it is divided in four main categories: primary, secondary, genetic, and of unknown causes [2]. Primary FSGS is also considered the most common cause of idiopathic nephrotic syndrome [3]. A more recent study has shown that although the incidence of FSGS is increasing, the ratio of primary and secondary FSGS has remained stable [4].

The etiology of primary FSGS has been attributed to a circulating factor, that is causing direct injury to the podocytes [5]. Damage to the podocytes is the initial event in the pathogenesis and diffuse foot process effacement is the earliest pathologic manifestation seen in the development of FSGS. The putative circulating factor, toxic to the podocyte, causes generalized podocyte dysfunction, manifested by widespread foot process effacement [6]. The identity of these factors has not yet been clearly established [7, 8]. Putative circulating permeability factors include the soluble form of the urokinase plasminogen activator receptor (suPAR) [9], cardiotrophin-like cytokine factor 1 (CLCF1) [10] and microRNAs [11, 12]. Remission of proteinuria is associated with improved renal survival as it has been shown in previous studies [13, 14]. Furthermore, it has been shown that even partial remission of proteinuria is associated with improved renal survival [15]. In the same study, it was shown that those who relapse from partial remission have worse renal survival compared to those that never relapse [15]. Spontaneous remission of primary FSGS is rare (<10%) in earlier studies from United States [16], while a more recent study from United Kingdom has shown spontaneous remission rate of 23% [14]. Prognosis is very poor for patients that do not go into remission as about 50% develop ESRD within 8 years [15, 16].

The treatment of primary FSGS is the use of immunosuppression, with high dose glucocorticoids being the first-line treatment [17]. Calcineurin inhibitors (CNIs) have also been studied for their efficacy in primary FSGS with best quality evidence coming from a few randomized trials in patients with steroid resistant primary FSGS where cyclosporine was shown to be superior to placebo regarding remission of proteinuria and preservation of renal function [18, 19]. The efficacy of cyclosporine has been studied as first line treatment only in a few retrospective studies, where CNIs were not shown to be superior to glucocorticoids regarding renal survival [20, 21].

The knowledge on FSGS and its causes continues to expand particularly regarding genetic causes of FSGS with more than 50 genes currently being known to be involved in FSGS [22]. This may help identify patients previously labelled as having primary FSGS, especially those who do not respond to current treatment available [23]. Despite these recent advances, primary FSGS continues to be a disease in which we still need further studies to guide clinicians in the treatment and long-term management.

Our study aimed at describing the clinical and laboratory characteristics of a large cohort of patients with primary FSGS, identify risk factors associated with worse renal survival and assess the impact of different immunosuppressive regiments on renal survival.

## Materials and methods

### Study design

This was a historical cohort study of adults who were diagnosed with primary FSGS from March 26, 1982 to September 16, 2020. The patients belonged to the Greek Registry of Primary FSGS, part of Glomerular Diseases Network. Twenty-one nephrology centers provided data for this study. The study was conducted according to the guidelines of the Declaration of Helsinki, and approved by the Institutional Review Board of all hospitals (IRB protocol number 22032024 for Laiko General Hospital, National and Kapodistrian University, Athens, Greece) and does not contain any identifiable patient data.

Diagnosis of Primary FSGS was based on renal biopsy findings. Clinical and laboratory parameters including clinical symptoms, medication, past medical history, laboratory results at time of diagnosis and routine investigation were analyzed (between 23/03/2024 to 28/04/2024) and reviewed. Patients with disorders strongly related with the secondary class of FSGS [disorders associated with reduced renal mass and renal vasolidation, drugs and toxins, as well as viral infections (particularly HIV-1, CMV, EBV)], or with family history of FSGS were excluded; only those who did not have any clear etiology for the adaptive FSGS were finally included in the study.

### Data collection

Baseline characteristics at the time of diagnosis including age, sex, BMI, degree of proteinuria (grams per 24 hours, based on 24-hour urine collection) were extracted from the patients' charts. Presence of arterial hypertension at the time of diagnosis (defined as systolic blood pressure ≥140 mmHg or diastolic blood pressure≥90mmHg) was also documented. We also documented treatment with Angiotensin–Converting Enzyme inhibitor (ACEi) or Angiotensin-Receptor Blocker (ARB) at the time of diagnosis. Complete Remission (CR) was defined as the reduction of 24-Hour urine protein levels to <0.3mg/d, normal serum albumin concentration, and normal eGFR. Partial Remission (PR) was described as the reduction of 24-hour urine protein levels by more than 50% from initial values but remaining 0.3–3.5g/d; accompanied by an improvement or normalization of the serum albumin concentration and stable eGFR. Finally, No Response (NR), was defined as 24-hour urine protein >3.5g/d or 0.3–3.5g/

d, but reduced less than 50% from initial values; and/or reduction of eGFR by >30%. Relapse of the disease was defined as the re-appearance of nephrotic syndrome after achieving complete or partial remission. The primary outcome was ESRD (dialysis or transplantation). eGFR was calculated using the CKD-EPI equation [24].

Immunosuppressive therapy was classified into three groups: glucocorticoids alone, cyclosporine with or without glucocorticoids, and other immunosuppressive agents (azathioprine, mycophenolate mofetil, Rituximab and cyclophosphamide).

End Stage Renal Disease (ESRD) was defined as the initiation of dialysis, or renal transplantation.

## Statistical analysis

Continuous variables were reported as mean ± standard deviation (SD) or median with interquartile ranges (IQR) and categorical variables were expressed as count (percent). P-values were derived from the t-test or Wilcoxon test for continuous variables and the Chi-square test or Fisher's exact test for categorical variables as appropriate.

The primary outcome was progression to ESRD. Kaplan-Meier curves were performed to assess the effect of remission status (CR vs. PR vs. NR) on progression to ESRD. Cox proportional hazards were performed to assess the association between each clinical and laboratory parameter collected and time to ESRD. Hazard Ratio (HR) was reported with the 95% confidence interval (CI). Cox proportional hazards assumption was tested using goodness of fit testing (Schoenfeld residuals). Multivariable analyses were performed to assess the effect of immunosuppression on progression to ESRD. Adjustments were made for the factors shown a statistical significance in the univariate analysis and for known potential confounders (age, sex, BMI). Baseline proteinuria and serum albumin concentrations were also included in the multivariate models, as they are major components in the clinical progression of the disease. The performance of a competing risk analysis, accounting for death as a competing risk on the progression to ESRD was not feasible, due to the limited availability on the data for the time to death in our cohort. However, for most of our patients, progression to ESRD preceded death. For the estimation of the Kaplan-Meier curves and the implementation of the Cox proportional hazards models, patients with available data on the time to death were censored at that time-point or when they were lost from follow-up prior to the development of ESRD. Statistical significance was determined when the p-value was <0.05. All analyses were performed using Stata 17 (StataCorp., College Station, TX).

## Results

Between March 26, 1982, and September 16, 2020, 699 patients were diagnosed with Primary FSGS. After excluding 28 patients that were less than 18 years of age and 92 patients that had follow up less than one year, 579 patients were included in the study (Fig 1).

The patient's baseline characteristics, follow-up, and outcomes are summarized in Table 1.

The mean age was 46 (±15) years, with 378 (65%) males, median 24-hour proteinuria was 3.8 (2–6) g and 337 (63%) of patients presented with hematuria. Of the 579 patients with FSGS that were included in the study, 309 received immunosuppression, 191 did not receive immunosuppression and in 79 patients there were not data available whether they received immunosuppression. Of those that received immunosuppression, 135 (23%) received glucocorticoids only, 126 (22%) patients received cyclosporine with or without glucocorticoids and 48 (8%) received other forms of immunosuppression (Rituximab, Mycophenolate, Cyclophosphamide, Azathioprine). The median duration of follow-up was 7.2 years (4–11.9). Complete remission

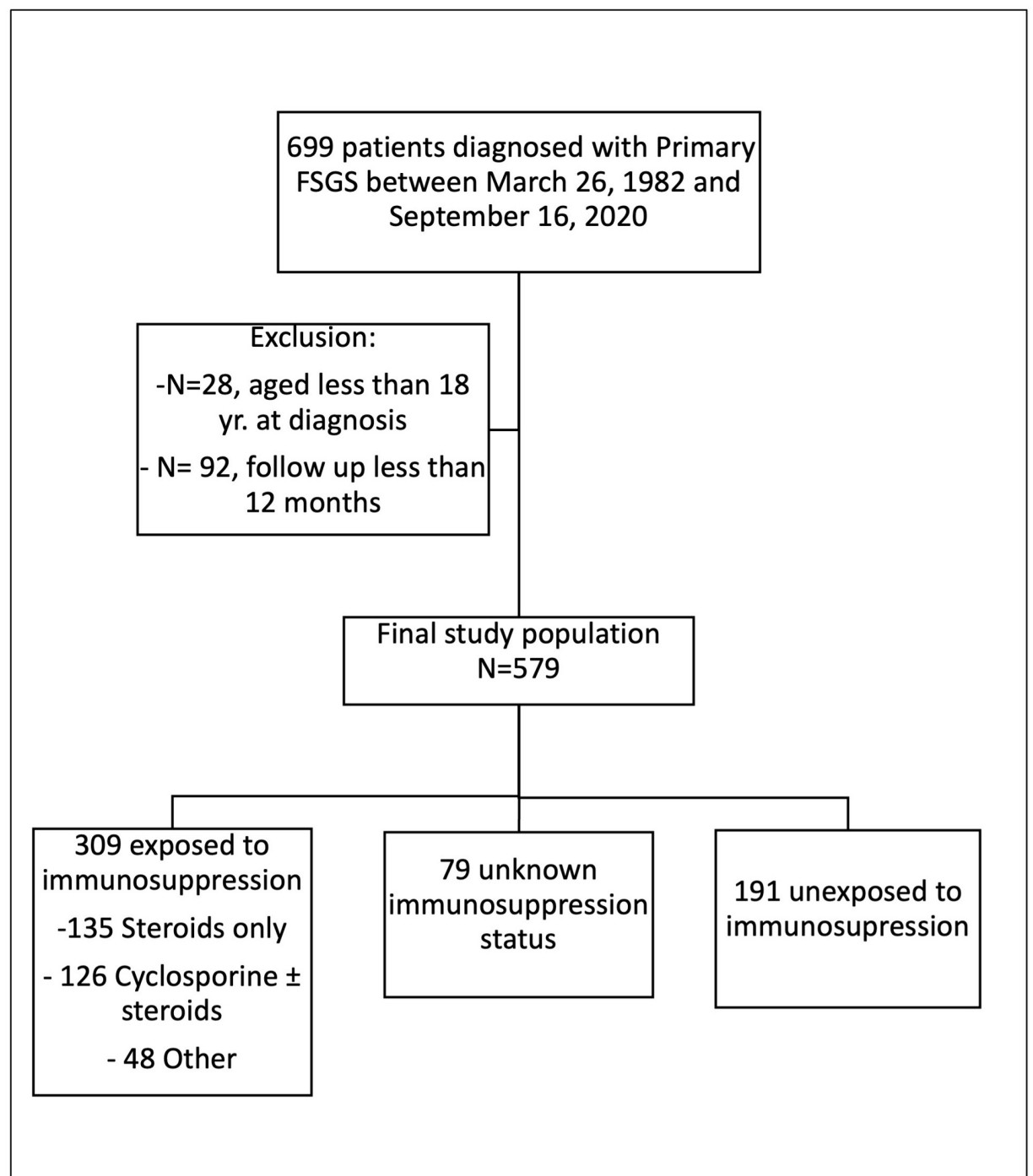

**Fig 1. Study population.** (Abbreviations: FSGS: Focal Segmental Glomerulosclerosis).

was achieved in 176 (35.5%) of patients, partial remission in 212 (43%) and no remission in 106 (22%).

When we compared the baseline characteristics between those that progressed to ESRD and those that did not progress, there was no statistical difference in age, sex, proteinuria at

**Table 1. Baseline characteristics of the cohort.**

| Baseline characteristic of 579 primary FSGS patients | |
| --- | --- |
| | N (%), Median (IQR), Mean (SD) |
| **Age (years)**[1] | 46 (±15) |
| **Sex (Males)** | 378 (65%) |
| **BMI (kg/m$^2$)**[2] | 28 (24–32.5) |
| **Proteinuria (g/d)**[2] | 3.8 (2–6) |
| **Microscopic Haematuria** | 337 (58%) |
| **Immunosuppression** | |
| • **Any form** | 309 (53%) |
| • **Glucocorticoids only** | 135 (23%) |
| • **Cyclosporine ± glucocorticoids** | 126 (22%) |
| • **Other** | 48 (8%) |
| **Hypertension** | 366 (63%) |
| **ACEi or ARB therapy (%)** | 396 (68%) |
| **Baseline eGFR (ml/min per 1.73 m$^2$)**[2] | 68 (44–94) |
| **Duration of follow up (years)**[1] | 7.2 (4–11.9) |
| **Outcomes** | |
| • **Remission (CR/PR/NR)** | 176 (35.5%) /212 (43%) / 106 (21.5%) |
| • **ESRD** | 88 (23%) |

[1]: Mean (SD),

[2]: Median (IQR)

BMI: Body Mass Index, ACEi: Angiotensin-converting enzyme inhibitors, ARB: Angiotensin Receptor Blockers, eGFR: estimated Glomerular Filtration Rate, CR: Complete Remission, PR: Partial Remission, NR: No Remission, ESRD: End Stage Renal Disease

diagnosis, serum albumin levels at diagnosis, treatment with ACEi or ARB at diagnosis or exposure to immunosuppression (S1 Table). However, eGFR at diagnosis was lower in those who progressed to ESRD compared to those who did not (43 (30–71.5) vs. 72 (51–96) ml/min per 1.73 m$^2$, p<0.001) and hypertension at diagnosis was more frequent in those that progressed to ESRD (67 (76%) vs. 200 (66%), p = 0.02). The percentage of patients that did not achieve remission was higher in those that progressed to ESRD (47 (53%) vs. 38 (13%), p<0.001). Relapse was more common in the patients that progressed to ESRD, compared to those that did not (24 (93%) vs. 109 (46%), p<0.001) and the number of relapses was also more common in the patients that progressed to ESRD (1 (1–2) vs. 0 (0–1), p<0.001).

We also compared the baseline characteristics between those that received immunosuppression and those that did not (Table 2).

There was no statistical difference regarding age, gender, BMI, baseline eGFR, presence of hypertension at diagnosis or progression to ESRD. However, proteinuria was higher in those that received immunosuppression compared to those that did not (4.91 (3.55–7.28) g/d vs. 2 (1.2–2) g/d, p<0.001), albumin was lower in those that received immunosuppression (3 (2.4–3.9) g/dL vs. 4 (3.6–4.3) g/dL, p<0.001), use of ACEi or ARB therapy at diagnosis was lower in those that received immunosuppression (220 (71%) vs. 165 (86%), p<0.001). Administration of immunosuppression showed correlation with remission status; among patients that received immunosuppression compared to those that did not there was a higher percentage of complete remission (121 (41%) vs. 40 (24%)), lower percentage of partial remission (125 (42%) vs. 79 (47%)) and lower percentage of no remission (52 (17%) vs. 50 (30%)), p<0.001. Finally,

**Table 2. Clinical and laboratory characteristics by immunosuppression status.**

| Immunosuppression Status | No Immunosuppression N = 191 | Immunosuppression N = 309 | p-value |
|---|---|---|---|
| | N (%), Median (IQR), Mean (SD) | | |
| Age (years)[1] | 47 (±12.9) | 45 (±17) | 0.25 |
| Sex (Males) | 128 (67%) | 199 (64%) | 0.55 |
| BMI (kg/m$^2$)[2] | 29 (25–33.6) | 27 (23.8–31.95) | 0.10 |
| Proteinuria (g/d)[2] | **2 (1.2–2)** | **4.91 (3.55–7.28)** | **<0.001** |
| Albumin (g/dL)[2] | **4 (3.6–4.3)** | **3 (2.4–3.9)** | **<0.001** |
| Baseline eGFR (ml/min per 1.73 m$^2$)[2] | 65 (44–93.5) | 68 (45–94) | 0.85 |
| Hypertension | 136 (71%) | 203 (66%) | 0.22 |
| ACEi or ARB therapy | **165 (86%)** | **220 (71%)** | **<0.001** |
| Remission | | | **<0.001** |
| • CR | **40 (24%)** | **121 (41%)** | |
| • PR | **79 (47%)** | **125 (42%)** | |
| • NR | **50 (30%)** | **52 (17%)** | |
| Relapse | **27 (33%)** | **135 (64%)** | **<0.001** |
| Number of Relapses[2] | **0 (0–1)** | **1 (0–2)** | **<0.001** |
| ESRD | 22 (17%) | 58 (23%) | 0.21 |

[1]: Mean (SD),

[2]: Median (IQR)

BMI: Body Mass Index, ACEi: Angiotensin-converting enzyme inhibitors, ARB: Angiotensin Receptor Blockers, eGFR: estimated Glomerular Filtration Rate, CR: Complete Remission, PR: Partial Remission, NR: No Remission, ESRD: End Stage Renal Disease

relapses were more common in the patients that received immunosuppression (135 (64%) vs. 27 (33%), P<0.001) and the number of relapses was higher in the patients that received immunosuppression (1 (0–2) vs. 0 (0–1), p<0.001).

In univariate Cox regression analysis, the parameters that were associated with increased likelihood of progression to ESRD included age (HR: 1.01, 95% CIs 0.99–1.02) and hypertension at diagnosis (HR: 2.11, 95% CIs 1.2–3.7), whereas baseline eGFR (HR: 0.97 per ml/min per 1.73 m$^2$ increase, 95% CIs 0.96–0.97) and treatment with ACEi or ARB at diagnosis (HR: 0.53, 95% CIs 0.33–0.86) were associated with decreased likelihood of progression to ESRD. Finally, remission status was also associated with progression to ESRD; complete remission (CR) (HR: 0.03, 95% CIs 0.01–0.1) and partial remission (PR) (HR: 0.18 95% CIs 0.10–0.31) were strongly correlated with decreased likelihood of progression to ESRD when compared to no remission (reference group) (Table 3).

We tried to further explore the association between proteinuria and the risk for ESRD, treating urine protein excretion as a continuous variable or categorical using different cut-offs. In fact, the results were comparable when proteinuria was treated as a continuous variable, or when we stratified it in proteinuric categories, or when we introduced heavy (>10g/day) urine protein excretion, compared to lower levels of proteinuria.

In Fig 2, the Kaplan—Meier curve of renal survival according to remission status (CR vs. PR vs. NR) is presented.

In multivariable analysis only baseline eGFR (HR 0.97 per ml/min increase, 95% CI 0.95–0.98) and remission status (complete remission (HR: 0.03, 95% CIs 0.003–0.22) and partial remission (HR: 0.28, 95% CIs 0.13–0.61) compared to no remission) were associated with renal survival (Table 3).

**Table 3. Cox regression model for predictors of time to ESRD.**

| Characteristics | Unadjusted HR (95% CIs) | p-value | Adjusted[a] HR (95% CIs) | p-value |
|---|---|---|---|---|
| Age, per year | 1.01 (0.99–1.02) | 0.20 | 0.98 (0.96–1.01) | 0.15 |
| Sex (Males) | 1.05 (0.66–1.65) | 0.37 | 0.94 (0.48–1.8) | 0.87 |
| BMI | 0.99 (0.97–1.01) | 0.53 | 0.99 (0.97–1.01) | 0.36 |
| Baseline proteinuria >3.5 g/d | 1.43 (0.91–2.2) | 0.11 | 0.90 (0.4–2) | 0.81 |
| Albumin (g/dl) | 0.83 (0.65–1.05) | 0.12 | 0.81 (0.53–1.2) | 0.35 |
| Baseline eGFR, ml/min | 0.97 (0.96–0.98) | <0.001 | 0.97 (0.95–0.98) | <0.001 |
| Hypertension | 2.11 (1.20–3.7) | 0.01 | 1.75 (0.79–3.8) | 0.16 |
| ACEi or ARB therapy | 0.53 (0.33–0.86) | 0.01 | 0.79 (0.37–1.67) | 0.54 |
| Immunosuppression | 1.58 (0.95–2.6) | 0.07 | 1.24 (0.58–2.6) | 0.56 |
| Remission | | | | |
| • NR | Reference Group | | | |
| • PR | 0.18 (0.10–0.31) | <0.001 | 0.28 (0.13–0.61) | 0.001 |
| • CR | 0.03 (0.01–0.1) | <0.001 | 0.03 (0.003–0.22) | 0.001 |

HR Hazard ratio, BMI: Body Mass Index, ACEi: Angiotensin-converting enzyme inhibitors, ARB: Angiotensin Receptor Blockers, eGFR: estimated Glomerular Filtration Rate, CR: Complete Remission, PR: Partial Remission, NR: No Remission

[a]Model adjusted for age, sex, hypertension at diagnosis, use of ACEi or ARB at diagnosis, BMI, degree of proteinuria at presentation (>3.5gr/ 24 HR), baseline eGFR and serum albumin

We also performed subgroup analysis in the patients that received immunosuppression; those that received glucocorticoids only vs. those that received calcineurin inhibitors ± glucocorticoids. We did not include patients that received other forms of immunosuppression, because most of the patients that were treated with immunosuppression received one of these two regiments (261/309, 84%) and the rest received several different regiments. There was no statistical difference regarding age, sex, degree of proteinuria, levels of serum albumin, baseline eGFR at diagnosis, presence of hypertension at diagnosis, remission status, relapse, or number of relapses between these two groups of patients. The only difference was more frequent use of ACEi or ARB therapy at diagnosis in the Cyclosporine group when compared to the group that received only glucocorticoids (106 (85%) vs. 80 (60%), p<0.001) (S2 Table).

In univariate Cox regression (Table 4), the parameters associated with better renal survival in this subgroup were baseline eGFR (HR: 0.96 per ml/min increase, 95% CIs 0.95–0.98) and treatment with ACEi or ARB at diagnosis (HR: 0.52 95% CIs 0.29–0.96), whereas presence of hypertension at diagnosis was associated with worse renal survival (HR: 3.36, 95% CIs 1.49–7.5). Remission status was also associated with progression to ESRD; complete remission (HR: 0.02, 95% CIs 0.01–0.09) and partial remission (HR: 0.13, 95% CIs 0.07–0.24) were strongly correlated with decreased likelihood of progression to ESRD, when compared to no remission. In multivariable analysis, only baseline eGFR (HR: 0.97 per ml/min per 1.73 m$^2$ increase, 95% CIs 0.95–0.99) and remission status (complete remission (HR: 0.03, 95% CIs 0.003–0.28) and partial remission (HR: 0.18, 95% CI 0.05–0.64) remained statistically significant as predictors of renal survival (Table 4). Similarly to the previous analysis on the whole cohort (Table 3), proteinuria, treated as a continuous variable or categorical using different cut-offs, showed no significant association with the time to ESRD.

In addition to the aforementioned models, we performed a sensitivity analysis including only the patients that had nephrotic range proteinuria (>3.5 g over 24 hours, N = 295

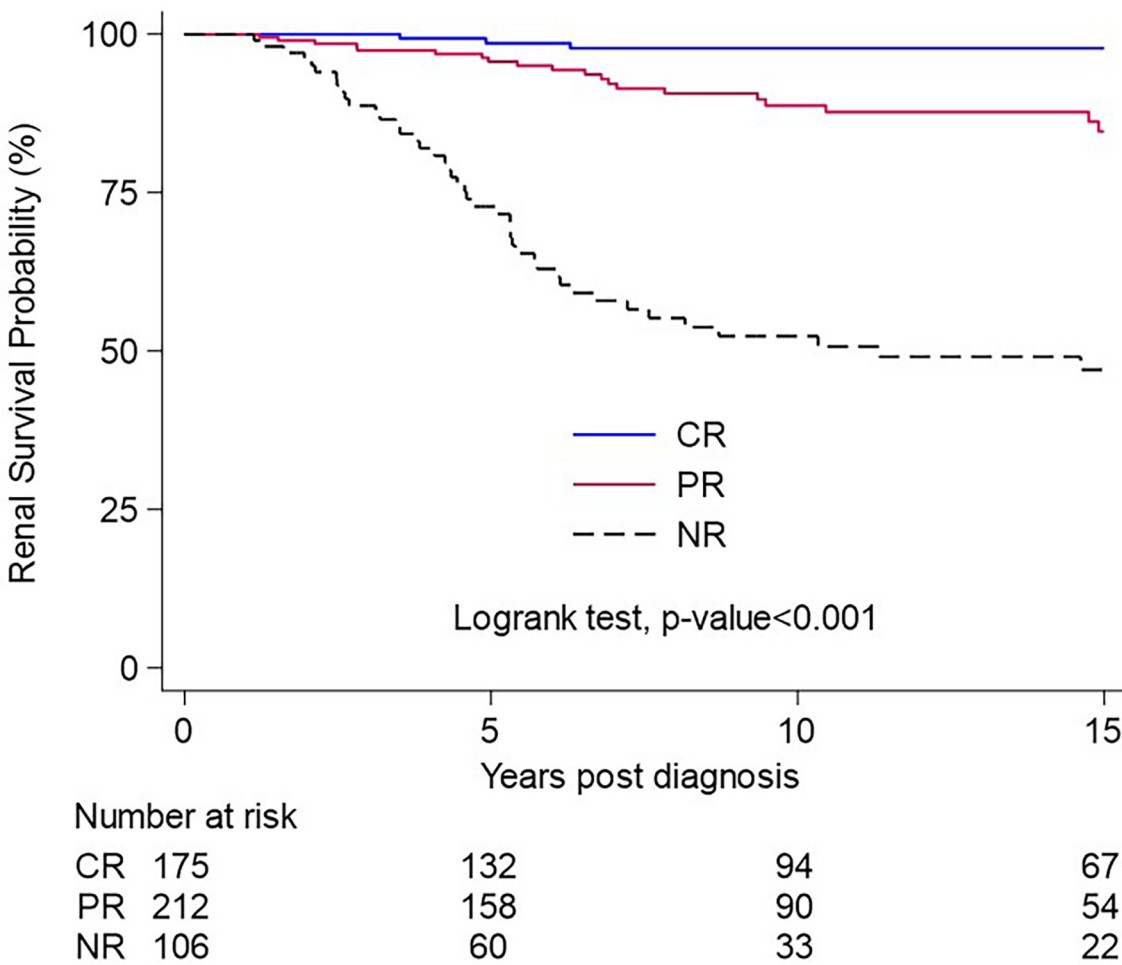

**Fig 2. Kaplan-Meier curve.** Survival from renal failure in patients with complete (CR), partial (PR), and no remission (NR).

patients). In multivariable analysis, only baseline eGFR (HR: 0.97 per ml/min per 1.73 $m^2$ increase, 95% CIs 0.95–0.99) and remission status (complete remission (HR: 0.02, 95% CIs 0.002–0.24) and partial remission (HR: 0.11, 95% CIs 0.02–0.47) compared to no remission) remained statistically significant predictors of renal survival. Exposure to immunosuppression was not associated with renal survival in univariate or multivariable analysis. We performed the same analysis among the subgroup of patients that received immunosuppression (glucocorticoids only vs. cyclosporine ± glucocorticoids). Results were similar with baseline eGFR, and remission status being associated with renal survival whereas type of immunosuppression (glucocorticoids only vs. Cyclosporine ± glucocorticoids) was not associated with renal survival.

Lastly, in order to explore any differences among the patients diagnosed with primary FSGS in different eras, we stratified our cohort based on the time of diagnosis; 113 (19.5%) had been diagnosed with FSGS before 2002, while almost 80% of the cohort had a more recent diagnosis (post 2002). The patients of the 2nd period were younger (47 vs 41 years of age, p-value = 0.002) and more likely to be males (67% vs 56.6% in the 1st period, p-value = 0.03). All other clinical and laboratory characteristics were comparable among patients of the two different periods (data not shown).

**Table 4. Cox regression model for predictors of time to ESRD, in the subgroup of patients that received immunosuppression (glucocorticoids only vs. cyclosporine ± glucocorticoids).**

| Characteristics | Unadjusted HR (95% CIs) | p-value | Adjusted[a] HR (95% CIs) | p-value |
|---|---|---|---|---|
| Age, per year | 1.01 (0.99–1.03) | 0.07 | 0.98 (0.95–1.01) | 0.273 |
| Sex (Males) | 1.37 (0.72–2.58) | 0.32 | 1.4 (0.49–4.1) | 0.5 |
| BMI (per kg/m$^2$) | 1 (0.97–1.02) | 0.98 | 1.01 (0.98–1.03) | 0.66 |
| Baseline proteinuria >3.5 g/d | 1.16 (0.56–2.4) | 0.67 | 1.63 (0.4–6.6) | 0.493 |
| Albumin (g/dl) | 0.91 (0.65–1.25) | 0.55 | 0.90 (0.49–1.66) | 0.758 |
| eGFR (baseline) ml/min per 1.73 m2 | **0.96 (0.95–0.98)** | **<0.001** | **0.97 (0.95–0.99)** | **<0.001** |
| Hypertension | **3.36 (1.49–7.5)** | **0.003** | 2.44 (0.57–10.29) | 0.223 |
| ACEi or ARB therapy | **0.52 (0.29–0.96)** | **0.038** | 0.66 (0.19–2.2) | 0.50 |
| Type of Immunosuppression | | | | |
| • Glucocorticoids only | Reference Group | | | |
| • Cyclosporine ± glucocorticoids | 0.70 (0.39–1.29) | 0.26 | 0.82 (0.30–2.1) | 0.693 |
| Remission | | | | |
| • NR | Reference Group | | | |
| • PR | **0.13 (0.07–0.24)** | **<0.001** | **0.18 (0.05–0.64)** | **0.009** |
| • CR | **0.02 (0.01–0.09)** | **<0.001** | **0.03 (0.003–0.28)** | **0.002** |

HR: Hazard ratio, BMI: Body Mass Index, ACEi: Angiotensin-converting enzyme inhibitors, ARB: Angiotensin Receptor Blockers, eGFR: estimated Glomerular Filtration Rate, CR: Complete Remission, PR: Partial Remission, NR: No Remission, ESRD: End Stage Renal Disease.

[a] Model adjusted for age, sex, hypertension at diagnosis, use of ACEi or ARB at diagnosis, BMI, degree of proteinuria at presentation (>3.5gr/ 24 HR), baseline eGFR and serum albumin

The achievement of response (either partial or complete) was comparable in the two periods (77.1% vs 78.4% in period 1 and 2, respectively, p-value = 0.78). In the estimation of the Kaplan-Meier curves (S1 Fig), patients diagnosed in the 1st period had a trend towards worse renal survival (Logrank test, p-value = 0.09), compared to the patients of the second period. In the multivariate analysis including the response status and the other significant confounders, however, no association was shown among the risk for ESRD and the period of diagnosis (adjusted HR: 0.81, p-value = 0.46).

## Discussion

In this large cohort of patients diagnosed with primary FSGS over a period of 38 years, we observed that higher eGFR at diagnosis and remission of proteinuria were associated with better renal survival in multivariable analysis. Complete and partial remission of proteinuria were associated with better renal survival compared to no remission. Patients that received immunosuppression had lower albumin and higher degree of proteinuria compared to those that did not receive immunosuppression. Also, the percentage of patients that achieved complete remission was higher in the group that received immunosuppression. Patients that received immunosuppression had a higher percentage of relapse and higher frequency of relapses. Immunosuppression was not associated with improved renal survival in univariate or multivariable analysis. In a subgroup analysis of patients that received glucocorticoids vs. cyclosporine ± glucocorticoids, baseline eGFR and remission status remained significant predictors of immunosuppression, while type of immunosuppression was not associated with better renal survival. Our findings were comparable in the analysis of the clinical progression and

risk for ESRD in patients diagnosed in different time-periods across the four decades of the study time-course.

Remission of proteinuria has been associated with better renal survival in several studies of patients with primary FSGS [14, 25, 26]. Partial remission has also been associated with renal survival in these patients in most recent studies [15, 27]. Our study confirmed this finding, as both complete and partial remission were associated independently with better renal survival both in univariate and multivariable analysis. Patients that did not achieve remission had a poor prognosis, with 50% of them reaching ESRD at 10 years after diagnosis, which is consistent with results of previous studies [14, 15]. Our findings support further the notion that complete and partial remission of proteinuria can be used as surrogate markers in future clinical trials [27]. A possible explanation for the lack of association between baseline proteinuria and the progression to ESRD is the inclusion of the response status in the multivariate models. It has been described that the response status, complete or even partial remission, is more important predictor (over baseline clinical and laboratory characteristics) in the long-term prognosis of patients with primary FSGS (8).

Higher baseline eGFR at diagnosis was associated with better renal survival, an expected finding which has been shown in previous studies [15, 21]. In univariate analysis, the use of ACEi or ARB was associated with better renal survival, while presence of hypertension at diagnosis was associated with worse renal survival. In multivariable analysis, after adjustments for remission of proteinuria, both the parameters lost their statistical significance, a finding that has been noted previously [15].

Immunosuppression was associated with a higher percentage of complete remission, but not with improved renal survival in our study. Older observational studies have shown that use of glucocorticoids was associated with higher likelihood of proteinuria remission [16, 25]. In our study patients received different immunosuppressive regimens with most of them receiving either glucocorticoids alone or cyclosporine ± glucocorticoids. As mentioned above, immunosuppression was not associated with renal survival, even though it was associated with complete remission of proteinuria in our study. Similar results were noted in a previous retrospective study where use of immunosuppression was associated with both partial and complete remission but not improved renal survival [15], whereas in a more recent study immunosuppression was associated with improved renal survival [21]. There are a few possible explanations for this finding; in our study patients that received immunosuppression had higher degree of proteinuria and lower albumin at baseline, suggesting that clinicians may have elected to give immunosuppression in patients with more severe clinical status and therefore, worse prognosis. Additionally, patients that received immunosuppression had higher rate of relapse, which may have had an impact on the effect of immunosuppression in preserving renal function.

We assessed the association of different immunosuppressive regiments (cyclosporine ± glucocorticoids vs. glucocorticoids only) and time to ESRD. To the best of our knowledge, this is the second study assessing the effect of CNIs in patients with primary FSGS as a first-line treatment [21]. In both studies there was not a statistical association between CNIs and better renal survival. Most of the evidence regarding CNIs in primary FSGS comes from trials that included patients with steroid-resistant nephrotic syndrome with the bulk of evidence showing that CNIs are more likely to induce partial or complete remission [28]. A randomized trial aiming to assess the efficacy of CNIs vs. glucocorticoids as first-line treatment in Primary FSGS may help clarify further their role in the treatment of this disease.

Our study has several limitations. First, we did not include histological parameters. Although specific FSGS variants have been associated with worse prognosis (collapsing FSGS) and others with better prognosis (tip lesion) in patients with primary FSGS [29], this finding

has not been consistent in all studies [26, 30]. Regardless, we were not able to extract data on the type of FSGS lesion and assess the association with renal survival. Consequently, we were not able to assess whether the degree of chronicity and fibrosis in the kidney biopsy had an impact not only in renal survival, but also on the administration of immunosuppression as clinicians may have been more reluctant to administer immunosuppression in patients with extensive fibrosis. Additionally, we did not have available data on time to remission nor on the duration on immunosuppressive therapy; therefore we could not assess factors associated with remission or the effect of the duration of treatment in multivariable analysis. Finally, although we did a thorough review of the medical records to include only patients with primary FSGS and the clinical characteristics of our patients indeed reflect the clinical status of other cohorts reporting outcomes in patients with primary FSGS [31], the inclusion of patients with a diagnosis almost 4 decades ago, before the important advancements in microscopy that have been made and the introduction of new types of FSGS, could have led to misclassification of some patients as Primary FSGS. There is a possibility that some patients with secondary FSGS, but more importantly with genetic causes of FSGS may have been included; for the latter we know they can present with clinical and laboratory characteristics very similar to Primary FSGS.

In conclusion, in this large cohort of patients with Primary FSGS, complete and partial remission of proteinuria were associated with better renal survival. Patients that received immunosuppression had more severe clinical presentation and the percentage of patients that achieved remission was higher compared to those that did not receive immunosuppression. However, administration of immunosuppression was not associated with improved renal survival, which can be partially explained by higher rate of relapse in those that received immunosuppression. CNIs were not shown to be superior to glucocorticoids regarding renal survival. Assessing the efficacy of CNIs and new therapeutic agents as first-line treatment of Primary FSGS in future randomized trials will provide further insight in the management of this disease.

## Supporting information

**S1 Table. Baseline characteristics by ESRD status.**
(DOCX)

**S2 Table. Clinical and laboratory parameters by immunosuppressive regimen.**
(DOCX)

**S1 Fig. Kaplan-Meier curves.** Survival from renal failure in patients with primary FSGS, according to the time of the diagnosis.
(DOCX)

## Author Contributions

**Conceptualization:** Smaragdi Marinaki, Panagiotis Kompotiatis, Ioannis Boletis.

**Data curation:** Smaragdi Marinaki, Panagiotis Kompotiatis, Ioannis Michelakis, Maria Stangou.

**Formal analysis:** Smaragdi Marinaki, Panagiotis Kompotiatis, Ioannis Michelakis.

**Investigation:** Smaragdi Marinaki, Panagiotis Kompotiatis, Maria Stangou, Aikaterini Papagianni, Maria Koukoulaki, Synodi Zerbala, Dimitrios Xydakis, Nikolaos Kaperonis, Evangelia Dounousi, Spyridon Golfinopoulos, Ioannis Stefanidis, Aggeliki Paikopoulou, George Moustakas, Kostas Stylianou, Ioannis Tzanakis, Marios Papasotiriou, Dimitrios Goumenos,

Aimilios Andrikos, Pelagia Kriki, Stylianos Panagoutsos, Eva Kiousi, Eirini Grapsa, Georgios Koutroumpas, Panagiotis Pateinakis, Dorothea Papadopoulou, Vasilios Liakopoulos, Dimitra Bacharaki, Penelope Kouki, Dimitrios Petras, Gerasimos Bamichas.

**Methodology:** Smaragdi Marinaki, Panagiotis Kompotiatis, Ioannis Michelakis.

**Project administration:** Smaragdi Marinaki, Ioannis Boletis.

**Resources:** Smaragdi Marinaki, Maria Stangou, Aikaterini Papagianni, Maria Koukoulaki, Synodi Zerbala, Dimitrios Xydakis, Nikolaos Kaperonis, Evangelia Dounousi, Spyridon Golfinopoulos, Ioannis Stefanidis, Aggeliki Paikopoulou, George Moustakas, Kostas Stylianou, Ioannis Tzanakis, Marios Papasotiriou, Dimitrios Goumenos, Aimilios Andrikos, Pelagia Kriki, Stylianos Panagoutsos, Eva Kiousi, Eirini Grapsa, Georgios Koutroumpas, Panagiotis Pateinakis, Dorothea Papadopoulou, Vasilios Liakopoulos, Dimitra Bacharaki, Penelope Kouki, Dimitrios Petras, Gerasimos Bamichas.

**Software:** Panagiotis Kompotiatis.

**Supervision:** Smaragdi Marinaki, Ioannis Boletis.

**Validation:** Smaragdi Marinaki, Panagiotis Kompotiatis, Ioannis Michelakis.

**Visualization:** Smaragdi Marinaki, Panagiotis Kompotiatis, Ioannis Michelakis.

**Writing – original draft:** Smaragdi Marinaki, Panagiotis Kompotiatis, Ioannis Michelakis.

**Writing – review & editing:** Smaragdi Marinaki, Panagiotis Kompotiatis, Ioannis Michelakis, Maria Stangou, Aikaterini Papagianni, Maria Koukoulaki, Synodi Zerbala, Dimitrios Xydakis, Nikolaos Kaperonis, Evangelia Dounousi, Spyridon Golfinopoulos, Ioannis Stefanidis, Aggeliki Paikopoulou, George Moustakas, Kostas Stylianou, Ioannis Tzanakis, Marios Papasotiriou, Dimitrios Goumenos, Aimilios Andrikos, Pelagia Kriki, Stylianos Panagoutsos, Eva Kiousi, Eirini Grapsa, Georgios Koutroumpas, Panagiotis Pateinakis, Dorothea Papadopoulou, Vasilios Liakopoulos, Dimitra Bacharaki, Penelope Kouki, Dimitrios Petras, Gerasimos Bamichas, Ioannis Boletis.

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
