## [Decision Letter · Decision Letter 0]

19 Aug 2024

PONE-D-24-17101Renal Survival and treatment of adult patients with Primary Focal Segmental Glomerulosclerosis: a historical cohort study of the National Greek RegistryPLOS ONE

Dear Dr. Michelakis,

Thank you for submitting your manuscript to PLOS ONE. After careful consideration, we feel that it has merit but does not fully meet PLOS ONE’s publication criteria as it currently stands. Therefore, we invite you to submit a revised version of the manuscript that addresses the points raised during the review process.

Please pay particular attention to the critiques from the two reviewers, including the concerns from Reviewer 1 about (1) distinction between primary and secondary FSGS, (2) if electron microscopic results were available for all subjects, and (3) if authors explored era effect in clinical outcomes; and concerns from Reviewer 2 on (1) competing risk analysis for the outcome of ESKD where death is treated as a competing risk, (2) if there are data on duration of therapy with immunosuppressants, and (3) using proteinuric categories in the model.    

We look forward to receiving your revised manuscript.

Kind regards,

Weining Lu, MD

Academic Editor

PLOS ONE

Journal Requirements:

Reviewers' comments:

Reviewer's Responses to Questions

**Comments to the Author**

1. Is the manuscript technically sound, and do the data support the conclusions?

Reviewer #1: Yes

Reviewer #2: Yes

2. Has the statistical analysis been performed appropriately and rigorously? 

Reviewer #1: Yes

Reviewer #2: Yes

3. Have the authors made all data underlying the findings in their manuscript fully available?

Reviewer #1: No

Reviewer #2: Yes

4. Is the manuscript presented in an intelligible fashion and written in standard English?

Reviewer #1: Yes

Reviewer #2: Yes

5. Review Comments to the Author

Reviewer #1: Dr. Michelakis and colleagues report on a historical cohort study evaluating more than 500 patients with presumed primary FSGS. This study and findings would be of interest to nephrology community.

Strengths:

- Large database for a relatively rare disease process

- Research aims to answer important clinical questions relating to kidney survival in patients with primary FSGS

I have few minor points that need additional clarification:

1. As distinction between primary and secondary FSGS is not always clear, were there standard adjudication practices in all 21 centers. If there was a central adjudication center, were information available from all sites qualitatively and quantitatively similar. I highly doubt that to be the case as the time course of this study spans 4 decades. How did authors adjudicate the exposure variable (Primary FSGS) if information available to them was not complete. While this limitation may not be fixable, it is important to better describe the exposure and outcome adjudication process, especially given the long timeframe of the study.

2. Were electron microscopic results available for all subjects? There needs to be some clarification on pathologic criteria of what denotes a primary FSGS vs. secondary FSGS. If EM was not available then how was diagnostic pathology used for disease identification?

2. Did authors explore era effect in clinical outcomes: For example, did people diagnosed in the last two decades fare better than the ones diagnosed in the first two decades? As treatment of proteinuric kidney disease (like use of renin-angiotensin-aldosterone blockers, calcineurin inhibitors) and treatment of vascular disease (like statins) have evolved significantly over the last 40 years, it would be of interest to know if there were differences in outcomes by era of practice.

3. Please be consistent in terms of reporting the number of decimal points in texts and tables. There is a mixer of one, two or more decimal points used. I suggest limiting everything to not more than 2 decimal points for consistency, except for when reporting low p-values.

Reviewer #2: In this paper, the authors investigated a large cohort of patients with primary FSGS to identify risk factors associated with renal survival and to assess the impact of immunosuppressive regiments on renal survival.

This is an interesting paper that is largely confirmatory of prior results but has long follow up data and a significant n of patients which are major strengths. I believe the following weaknesses should be addressed:

Major points:

Due to the long duration of follow up, data on death are also needed. The authors should conduct a competing risk analysis for the outcome of ESKD where death is treated as a competing risk.

It’s interesting that proteinuria is not significant in the fully-adjusted model, might it be worthwhile to explore proteinuria not as a dichotomized variable (>/< 3.5) but in proteinuric categories in these models?

Are there data on duration of therapy with immunosuppressives that can be included?

Minor points:

Introduction line 68: would benefit from more detail here, what is this/are these potentially podocyte-toxic factors etc.

Line 123 can you specify causes for sec fsgs and your workflow for excluding patients

What was the rationale for not using CKD EPI to calculate eGFR?

In the methods section, please list confounders and exact adjustment strategy (in addition to table legend). This will be easier for the reader to follow

Please comment on when patients were censored, what happened when they died?

Please use same nomenclature in text and table for (SD) and [IQR]

Please add units and what you measure throughout whether it is a ratio or a percentage etc (e.g., line 179,180)

6. PLOS authors have the option to publish the peer review history of their article (what does this mean?). If published, this will include your full peer review and any attached files.

Reviewer #1: No

Reviewer #2: No

---

## [Author Response · Author response to Decision Letter 0]

15 Oct 2024

Response to reviewers’ comments

ID: PONE-D-24-17101

Title: Renal Survival and treatment of adult patients with Primary Focal Segmental Glomerulosclerosis: a historical cohort study of the National Greek Registry

Dear Editor,

We would like to thank the reviewers for their insightful comments and suggestions that helped to improve our manuscript.

Please see below a point-by-point responses to these comments. Our updated manuscript with tracked modifications has also been uploaded, as long as an unmarked version of the revised paper. 

We have uploaded a revised version of the Supplementary Materials, and a revised file of the Figure 2. 

Reviewer #1

Dr. Michelakis and colleagues report on a historical cohort study evaluating more than 500 patients with presumed primary FSGS. This study and findings would be of interest to nephrology community.

Strengths:

- Large database for a relatively rare disease process

- Research aims to answer important clinical questions relating to kidney survival in patients with primary FSGS

I have few minor points that need additional clarification:

1. As distinction between primary and secondary FSGS is not always clear, were there standard adjudication practices in all 21 centers. If there was a central adjudication center, were information available from all sites qualitatively and quantitatively similar. I highly doubt that to be the case as the time course of this study spans 4 decades. How did authors adjudicate the exposure variable (Primary FSGS) if information available to them was not complete. While this limitation may not be fixable, it is important to better describe the exposure and outcome adjudication process, especially given the long timeframe of the study.

Authors’ Response

Thank you for your insightful feedback on our manuscript. We appreciate your concerns regarding the practice for the diagnosis of patients with FSGS and their inclusion in the study. 

The central adjudication center for this study was the Department of Nephrology of Laiko General Hospital of Athens, which was responsible for the Greek Registry of Primary FSGS, part of Glomerular Diseases Network. The diagnosis of Primary FSGS was based on renal biopsy findings and there was a consensus over the inclusion criteria of each patient in the study. Moreover, a common electronic chart for all the centers was made prior to the implementation of the study, with clear, common instructions on the way that it should be completed. 

Most importantly, the medical records of all the patients and the biopsy reports were carefully re-evaluated for the purposes of this study from all centers, in order to minimize the inclusion of potential non-primary cases of FSGS. 

The clinical characteristics of the patients in our study indeed reflect the clinical status of other cohorts reporting outcomes in patients with primary FSGS (1). We do recognize, however, that the inclusion of patients with a diagnosis almost 4 decades ago, before the important advancements in microscopy that have been made and the introduction of new types of FSGS, could have led to misclassification of some patients as Primary FSGS. We have pointed out this important limitation in the Discussion to emphasize further on the issue after your comments. 

As far as the outcomes measured is concerned, only hard outcomes have been recorded and analyzed in the study. We evaluated the response status (complete, partial or no remission), the incidence of relapses, the progression to end-stage renal disease (ESRD) and the mortality of our patients. For this reason, we believe that, given the large number of patients included, our findings provide useful insight on the clinical progression of patients with Primary FSGS, with a potential misclassification bias, which was unavoidable, but minimum.

2. Were electron microscopic results available for all subjects? There needs to be some clarification on pathologic criteria of what denotes a primary FSGS vs. secondary FSGS. If EM was not available then how was diagnostic pathology used for disease identification?

Authors’ Response

We appreciate your comment and understand the importance of electron microscopy (EM) data in providing detailed insights into the pathology of kidney disease. However, EM reports were not available in this study, mainly due to limited access in all the centers. Indeed, there is an increasing appreciation that electron microscopy is particularly useful in the diagnosis of genetic variant of FSGS (2), as they may produce characteristic changes within the glomerular basement membrane. We acknowledge that the absence of EM findings is a limitation, and for this reason we have addressed the minor use of microscopy findings to the limitations’ section of our manuscript. Despite this, we believe that the other diagnostic methods employed (careful re-assessment of the light microscopy reports and patients’ medical records) on a very large cohort of patients with presumable primary FSGS provide sufficient information to support our conclusions.

3. Did authors explore era effect in clinical outcomes: For example, did people diagnosed in the last two decades fare better than the ones diagnosed in the first two decades? As treatment of proteinuric kidney disease (like use of renin-angiotensin-aldosterone blockers, calcineurin inhibitors) and treatment of vascular disease (like statins) have evolved significantly over the last 40 years, it would be of interest to know if there were differences in outcomes by era of practice.

Authors’ Response

We would like to thank you for your insightful comment. Following your suggestion, we stratified our cohort based on the time of diagnosis; 113 (19.5%) had been diagnosed with FSGS before 2002, while almost 80% of the cohort had a more recent diagnosis (post 2002). The patients of the 2nd period were younger (47 vs 41 years of age, p-value=0.002) and more likely to be males (67% vs 56.6% in the 1st period, p-value=0.03). All other clinical and laboratory characteristics were comparable among patients of the two different periods.

The achievement of response (either partial or complete) was comparable in the two periods (77.1% vs 78.4% in period 1 and 2, respectively, p-value=0.78). 

In the estimation of the Kaplan-Meier curve (Figure 1), patients diagnosed in the 1st period had a trend towards worse renal survival (Logrank test, p-value=0.09), compared to the patients of the second period. In the multivariate analysis including the response status and the other significant confounders, however, no association was shown among the risk for ESRD and the period of diagnosis (adjusted HR: 0.81, p-value=0.46).

Figure 1. Kaplan-Meier Curves. Survival from renal failure in patients with primary FSGS, according to the time of the diagnosis.

We have added a description of the patients according to the time of FSGS diagnosis and the analysis performed in the Results Section. 

4. Please be consistent in terms of reporting the number of decimal points in texts and tables. There is a mixer of one, two or more decimal points used. I suggest limiting everything to not more than 2 decimal points for consistency, except for when reporting low p-values.

Authors’ Response

Thank you for your suggestion. We have now revised our manuscript and present the reported numbers in a consistent way.

Reviewer #2

In this paper, the authors investigated a large cohort of patients with primary FSGS to identify risk factors associated with renal survival and to assess the impact of immunosuppressive regiments on renal survival.

This is an interesting paper that is largely confirmatory of prior results but has long follow up data and a significant n of patients which are major strengths. I believe the following weaknesses should be addressed:

Major points:

1. Due to the long duration of follow up, data on death are also needed. The authors should conduct a competing risk analysis for the outcome of ESKD where death is treated as a competing risk.

Authors’ Response

Thank you very much for pointing out the need for integrating data on death in our manuscript. Our intention was to perform a competing risk analysis, accounting for death as a competing risk on the progression to ESRD. However, the availability on the data for the time to death was limited for our cohort. For many of our patients we were able to retrieve information on the incidence of death, but not on the time to death. We re-assessed our data after your comment; For most of our patients, progression to ESRD preceded death. We were able to perform a competing risk analysis, including data on time to death for 20 patients; they all died with preserved function. The results from this analysis are comparable to those presented in the main manuscript. We have revised the Methods section accordingly.

2. It’s interesting that proteinuria is not significant in the fully-adjusted model, might it be worthwhile to explore proteinuria not as a dichotomized variable (>/< 3.5) but in proteinuric categories in these models?

Authors’ Response

Thank you very much for the careful interpretation of our manuscript and the results. Indeed, the absence of a significant effect of nephrotic levels of proteinuria at baseline on ESRD progression was an interesting finding. We tried to further explore the association between proteinuria and the risk for ESRD and performed the models, treating urine protein excretion in different ways. In fact, the results were comparable when proteinuria was treated as a continuous variable, or when we stratified it in proteinuric categories; even when we introduced a dummy variable of heavy (>10g/day) urine protein excretion, compared to lower levels of proteinuria. 

A possible explanation in the inclusion of the response status in the multivariate model. It has been described that the response status, complete or even partial remission, is more important predictor (over baseline clinical and laboratory characteristics) in the long-term prognosis of patients with primary FSGS (3). 

We have commented on the findings in the results section and commented on that in the Discussion. 

3. Are there data on duration of therapy with immunosuppressives that can be included?

Authors’ Response

Thank you very much for your question. We do recognize that the duration of immunosuppressive therapy could have an impact on the observed outcomes, both in the response status, but also in the risk for flare and long-term progression of the patients included in the study. However, only the type of immunosuppressants was known, and not the duration of treatment. Given the large size of our cohort and the focus on well-described outcomes, we believe that the omission of treatment duration data is unlikely to have significantly impacted our findings. The robustness of the sample size and the consistency of the outcomes measured likely mitigates any potential bias related to treatment duration variability. For clarity, we have pointed this limitation in the Discussion section. 

Minor points:

1. Introduction line 68: would benefit from more detail here, what is this/are these potentially podocyte-toxic factors etc.

Authors’ Response

Thank you very much for your suggestion. We have now further elaborated in this specific section of the introduction. 

«Damage to the podocytes is the initial event in the pathogenesis and diffuse foot process effacement is the earliest pathologic manifestation seen in the development of FSGS. A putative circulating factor, toxic to the podocyte, causes generalized podocyte dysfunction, manifested by widespread foot process effacement (4). The identity of these factors has not yet been clearly established (5, 6). Putative circulating permeability factors include the soluble form of the urokinase plasminogen activator receptor (suPAR) (7), cardiotrophin-like cytokine factor 1 (CLCF1) (8) and microRNAs (9, 10).»

2. Line 123 can you specify causes for sec fsgs and your workflow for excluding patients

Authors’ Response

We appreciate your suggestion. The causes of Secondary FSGS and our workflow has been elaborated in the Study Design section in Methodology.

“Patients with disorders strongly related with the secondary class of FSGS [disorders associated with reduced renal mass and renal vasolidation, drugs and toxins, as well as viral infections (particularly HIV-1, CMV, EBV)], or with family history of FSGS were excluded; only those who did not have any clear etiology for the adaptive FSGS were finally included in the study.”

3. What was the rationale for not using CKD EPI to calculate eGFR?

Authors’ Response

Thank you very much for your question. There was an error in the preparation of the manuscript; the CKD-EPI equation, and not MDRD was used to calculate eGFR in our analysis. In the following link you may find our data with date of publication (Jan 8th, 2024) prior to the submission or our manuscript. 

https://data.mendeley.com/datasets/9mjn55c3x9/1

As you may see in the dataset, the name of the column with the eGFR data includes also CKD-EPI as a label. We have corrected this error in the manuscript.

4. In the methods section, please list confounders and exact adjustment strategy (in addition to table legend). This will be easier for the reader to follow

Authors’ Response

We would like to thank you for your comment. The exact methodology for the variables included in the multivariate models performed in our study has now been described in the Statistical Analysis section of the Methodology.

“Multivariable analyses were performed to assess the effect of immunosuppression on progression to ESRD. Adjustments were made for the factors shown a statistical significance in the univariate analysis and for known potential confounders (age, sex, BMI). Baseline proteinuria and serum albumin concentrations were also included in the multivariate models, as they are major components in the clinical progression of the disease.”

5. Please comment on when patients were censored, what happened when they died?

Authors’ Response

We appreciate your comment on the censoring of our data in the analysis of the risk factors for progression to ESRD. For the estimation of the Kaplan-Meier curves and the implementation of the Cox proportional hazards models, patients with available data on the time to death were censored at that time-point or when they were lost from follow-up prior to the development of ESRD. As previously described on your comment on the decision not to perform a competing risk analysis, accounting for death as a competing risk on the progression to ESRD, the availability on the data for the time to death was limited for our cohort. We tried to carefully re-assess our data, and for the majority of our patients, progression to ESRD preceded death. For this reason, we focused the description of our findings on investigating the determinants and risk factors of progression to ESRD, as, with our data, we would not be able to elaborate on the risk for mortality in patients with primary FSGS. 

We have described the time when patients were censored in the Methodology section.

6. Please use same nomenclature in text and table for (SD) and [IQR]

Authors’ Response

Thank you for your comment. We have revised our manuscript and present the reported numbers in a similar and consistent way.

7. Please add units and what you measure throughout whether it is a ratio or a percentage etc (e.g., line 179,180)

Authors’ Response

Thank you for your comment. We have revised the presentation of the results in the section you addressed. 

References

1. Stamellou, E., Nadal, J., Hendry, B., Mercer, A., Bechtel-Walz, W., Schiffer, M., Eckardt, K. U., Kramann, R., Moeller, M. J., Floege, J., & GCKD study investigators (2024). Long-term outcomes of adults with FSGS in the German Chronic Kidney Disease cohort. Clinical kidney journal, 17(7), sfae131. https://doi.org/10.1093/ckj/sfae131

2. Davis, J., Tjipto, A., Hegerty, K., & Mallett, A. (2019). The use of electron microscopy in the diagnosis of focal segmental glomerulosclerosis: are current pathological techniques missing important abnormalities in the glomerular basement membrane?. F1000

---

## [Decision Letter · Decision Letter 1]

21 Nov 2024

Renal Survival and treatment of adult patients with Primary Focal Segmental Glomerulosclerosis: a historical cohort study of the National Greek Registry

PONE-D-24-17101R1

Dear Dr. Michelakis,

We’re pleased to inform you that your manuscript has been judged scientifically suitable for publication and will be formally accepted for publication once it meets all outstanding technical requirements.

Kind regards,

Weining Lu, MD

Academic Editor

PLOS ONE

Additional Editor Comments (optional):

Reviewers' comments:

Reviewer's Responses to Questions

**Comments to the Author**

1. If the authors have adequately addressed your comments raised in a previous round of review and you feel that this manuscript is now acceptable for publication, you may indicate that here to bypass the “Comments to the Author” section, enter your conflict of interest statement in the “Confidential to Editor” section, and submit your "Accept" recommendation.

Reviewer #1: All comments have been addressed

Reviewer #2: All comments have been addressed

2. Is the manuscript technically sound, and do the data support the conclusions?

Reviewer #1: Yes

Reviewer #2: Yes

3. Has the statistical analysis been performed appropriately and rigorously? 

Reviewer #1: Yes

Reviewer #2: Yes

4. Have the authors made all data underlying the findings in their manuscript fully available?

Reviewer #1: Yes

Reviewer #2: Yes

5. Is the manuscript presented in an intelligible fashion and written in standard English?

Reviewer #1: Yes

Reviewer #2: Yes

6. Review Comments to the Author

Reviewer #1: Authors have addressed all my concerns during their revision. The manuscript is at an acceptable level for publication and will be of an interest to nephrology community.

Reviewer #2: (No Response)

7. PLOS authors have the option to publish the peer review history of their article (what does this mean?). If published, this will include your full peer review and any attached files.

Reviewer #1: **Yes: **Ashish Upadhyay

Reviewer #2: No

---

## [Editor Report · Acceptance letter]

2 Dec 2024

PONE-D-24-17101R1 

PLOS ONE

Dear Dr. Michelakis, 

I'm pleased to inform you that your manuscript has been deemed suitable for publication in PLOS ONE. Congratulations! Your manuscript is now being handed over to our production team.

Kind regards, 

on behalf of

Dr. Weining Lu 

Academic Editor

PLOS ONE